# Graph-based approaches for Hybrid AI solutions

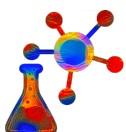

TL;DR: *Integration of several disparate graph-based technologies available as Python open source, made to work efficiently with popular tooling for Data Science and Data Engineering practices, leading toward Hybrid AI solutions*

Python offers excellent libraries for working with graphs: semantic technologies, graph queries, interactive visualizations, graph algorithms, probabilistic graph inference, as well as embedding and other integrations with deep learning. However, most of these approaches share little common ground, nor do many of them integrate effectively with popular data science tools (pandas, scikit-learn, spaCy, PyTorch), nor efficiently with popular data engineering infrastructure such as Spark, RAPIDS, Ray, Parquet, fsspect, etc. The library has use cases in large enterprise firms in industry and is also used as a teaching tool.

This talk reviews **kglab** – an open source project focused on integrating the priorities described above, and moreover providing ways to leverage disparate graph technologies in ways that complement each other, to produce Hybrid AI solutions for industry use cases. At its core, this effort is about self-supervised learning in graph-based data science workflows, leading toward Hybrid AI solutions. We'll cover some of the less intuitive learnings which have provided practical guidance in this work. For example, the notion of "Thinking Sparse and Dense", to make the most of available subsystems, in software and hardware respectively, when working with graph data. Similarly, how transforms and inverse transforms based on algebraic graph theory apply for effective design patterns in this integration work. We'll also consider when to make trade-offs between more analytic methods versus tools that allow for uncertainty in the data, and also how to blend data-intensive machine learning with rule systems based on domain expertise.

keywords: *knowledge graph*, *data science*, *graph algorithms*, *probabilistic graph inference*, *deep learning*, *hardware accelerators*, *hybrid ai*, *artificial intelligence*, *design patterns*

curated public superset of slides: **https://derwen.ai/s/kcgh**

public repository: **https://github.com/DerwenAI/kglab**

DOI: **10.5281/zenodo.4602403**

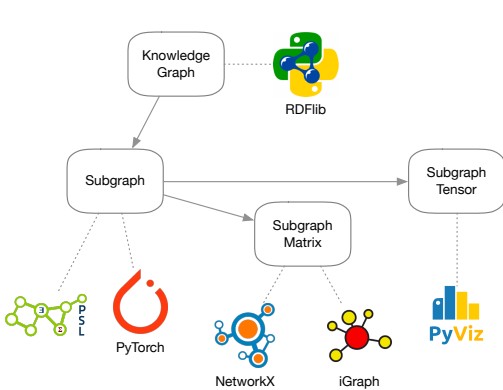

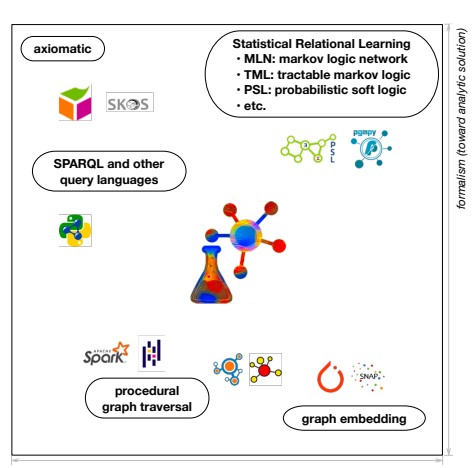