# OpenReview forum: "Graph-based approaches for Hybrid AI solutions"
_vivoconference.org/VIVO/2021/Conference_

### Official Review · Program_Chairs · 2021-05-29
**What can we learn from graphs of scholarship?**

**Rating:** 9
**Confidence:** 4

**Review:**

VIVO provides the means for creating open, detailed, graphs of scholarship across the world -- how the works are created, variety of works beyond papers, including projects, patents, "grey literature", early results, proposals, and other indicators of research interest.

The proposed talk provides an overview of technologies useful in analyzing graph-based data using AI techniques.  The VIVO audience is likely to be mostly unaware of these techniques and their predecessors.  An introductory talk regarding the value of such technqiues for inference from graph-based data would be attractive to the VIVO community,

Insights regarding the use of the techniques to answer questions about scholarship posed by academic leaders and those interested in the the current and future research activities of the academic community would be most appreciated.